# Association between food insecurity, ethnicity, and mental health in the UK: An analysis of the Family Resource Survey

Maddy Power[1]*, Tiffany Yang[2], Katie Pybus[3], Benham Tajik[1]

1 Department of Health Sciences, University of York, Heslington, York, United Kingdom, 2 Bradford Institute for Health Research, Bradford Teaching Hospitals NHS Foundation Trust, Bradford, United Kingdom, 3 Hull York Medical School, York, United Kingdom

* madeleine.power@york.ac.uk

## Abstract

The study aimed to assess the relationship between food insecurity and ethnicity in the United Kingdom (UK), and to explore how the relationship between food insecurity and mental health varies by ethnic group. Data from the 2019/20 Family Resource Survey provided information on ethnicity, presence of long-standing illnesses affecting mental health, and food security assessed using 10-item Adult Food Security module. Logistic regression was used to assess the relationship between food security status and degree of anxiety and presence of long-standing illness affecting mental health. Analyses were adjusted for covariates and stratified by ethnicity. Participants were a representative sample of private UK households (N = 19,210), using the Household Reference Person as the main respondent. The majority of the sample were food secure (87%), identified as White (90.7%), and 22% reported a long-standing illness affecting mental health. Food insecurity was associated with longstanding illness affecting mental health (adjusted OR 2.01 (1.70, 2.39)) among all ethnic groups; Asian/Asian British respondents reported the highest odds of having a longstanding illness affecting their mental health (OR=2.63 (1.05, 6.56)). The study finding of an association between food insecurity and mental health for all UK ethnic groups, but one which is stronger for ethnic minority groups, necessitates a population-wide response alongside targeted interventions.

## Introduction

Over the last decade, food insecurity, defined as 'the limited or uncertain availability of nutritionally adequate and safe foods or limited or uncertain ability to acquire acceptable foods in socially acceptable ways' [1], has become increasingly relevant to the health of United Kingdom (UK) populations. According to the Department for Work and Pensions' (DWP) Households Below Average Income survey, in 2023/24,

**Data availability statement:** The data is fully available from the Department for Work and Pensions (UK), at: https://www.gov.uk/government/collections/family-resources-survey--2.

**Funding:** This research was funded by a Wellcome Trust Research Fellowship in Humanities and Social Sciences held by Maddy Power. Grant Number: 221021/Z/20/Z The funders had no role in study design, data collection and analysis, decision to publish, or preparation of the manuscript.

**Competing interests:** The authors have declared that no competing interests exist.

10% of UK households were food insecure with an additional 7% reporting marginal food security [2].

The sharp increase in food prices since the second half of 2021 has directly impacted the food security of UK households. In July and August 2023, 56% of adults in Great Britain reported an increase in their cost of living compared with the month before [3]. Of these, 97% saw the price of their food shopping go up, and 47% had started spending less on essentials including food. The rising cost of living appears to be increasing household food insecurity. A YouGov survey by the Food Foundation, a food poverty charity, found that in June 2023, 17% of households in the UK were 'food insecure' (ate less or went a day without eating because they couldn't access or afford food), up from 8.8% in January 2022 and 7.4% in January 2021 [4].

Food insecurity in the UK today is arguably a public health crisis but one that remains only partially understood. Emerging evidence points towards a relationship with mental health [5] and shows that food bank usage, is particularly prevalent among individuals who report mental health problems [2]. Albeit there is also academic evidence to suggest that, in replicate stigmatising welfare state bureaucracy, food banks may also contribute to mental ill-health [3].

Like food insecurity, mental health is increasingly becoming an urgent and widespread public health concern in the UK and worldwide. According to the World Health Organization, a mental disorder is characterized by a clinically significant disturbance in an individual's cognition, emotional regulation, or behaviour [4]. It is usually associated with distress or impairment in important areas of functioning. There are many different types of mental disorders. Mental disorders may also be referred to as mental health conditions. The latter is a broader term covering mental disorders, psychosocial disabilities and (other) mental states associated with significant distress, impairment in functioning, or risk of self-harm [4]. In 2019, 1 in every 8 people, or 970 million people around the world were living with a mental disorder, with anxiety and depressive disorders the most common [5].

The relationship between mental health and food insecurity, identified in the UK, is in line with the well-established association between mental health and food insecurity in North America [6]. For instance, Canadians living in food insecure households are at greater risk of poor mental health than those living in food secure households and this risk increases with the severity of food insecurity [7,8]. Adults living in food insecure households are more likely to experience a wide range of adverse mental health outcomes, compared to those living in food secure households, including depressive and suicidal thoughts, major depressive episodes, diagnosed mood disorders, diagnosed anxiety disorders and self-reported poor mental health [8–10]. Evidence suggests that stress plays a key role in the association between food insecurity and mental health; existing mental health issues, in particular, are exacerbated by poor access to food leading to a cycle of stress and deteriorating mental health [11].

Food insecurity is not spread evenly across the UK population with income, age, disability, geography and ethnicity affecting household food insecurity [12,13]. The relationship between ethnicity and food insecurity remains relatively neglected in literature on UK food insecurity, an omission which may be related to the historical

absence of routinely collected national-level data on food insecurity, prior to the publication of the 2019−20 Family Resource Survey (FRS) (in 2021) [14]. Descriptive statistics in the FRS show high food insecurity among Black/African/ Caribbean/Black British households and comparatively low food insecurity among White and Indian households [14]. Analysis of the nationally representative 2016 Food and You survey [15] similarly illustrated differences in food insecurity by ethnicity. The relatively small size of the survey forces the researchers to use reductive categories of ethnic group: 'white' and 'Other ethnic group'/ 'non-white'. Using these categories, their analysis suggests that adults who describe themselves as belonging to a 'non-white' ethnic group are more likely to report food insecurity than adults identifying as 'white'. The researchers also find that ethnicity is associated with moderate food insecurity but not with severe food insecurity, suggesting that ethnicity may be a less important factor in food insecurity than other characteristics, such as income, health status and gender.

At a local level, however, the picture may be more complex. Analysis of food insecurity in the Born in Bradford survey (including pregnant women living in the Bradford District) found that Pakistani British women were less likely than white British women to report food insecurity [16]. Assessment of food bank usage by ethnic group has also identified an under-representation of some ethnic minorities [13,17]. Research suggests that there may be multiple reasons for the potential under-representation of some ethnic minority groups at food banks, including less awareness, for instance among South Asian compared to White British populations, of food banks and related food aid services; cultural barriers to seeking support from food banks, for example stigma and shame surrounding food insecurity among South Asian populations in Bradford [18]; and experiences of discrimination within food banks alongside the absence of culturally appropriate food at food banks, and in low income areas more broadly [19,20]. Some research has suggested that kinship networks among South Asian communities may mitigate food insecurity [21].

The apparent raised risk of ethnic minority groups to food insecurity and the established link between food insecurity and poor mental health suggests that ethnic minority groups may be at higher risk of food insecurity-related poor mental health. Empirical evidence on the relationship between food insecurity and mental health among different ethnic groups is limited, especially in the UK. Current evidence in North America indicates that food insecurity is associated with varying levels of serious psychological distress among minority ethnic populations and among Black/African Americans specifically being food insecure with hunger contributes to experiencing serious psychological distress [22–25]. In the UK, evidence suggests a stronger relationship between food insecurity and common mental health disorders among White British compared to Pakistani pregnant women [26] but also points towards a stigma surrounding reporting mental ill health among some minority ethnic groups [27].

In the UK, people from ethnic minority groups are more likely to have undiagnosed and untreated mental illness [28–30], enter healthcare via crisis or other aversive pathways [31], and receive a diagnosis of severe mental illness compared to the majority ethnic group [32,33]. People in ethnic minority households are more likely to be in persistent low income before and after housing costs than people in white households [33], a factor that has been positively correlated with greater rates of mental illness (including long-standing mental illness) and food insecurity [34–36].

Living on a low income has well-established links with poor mental health [37] because the broader experience of poverty is often stressful [38]. In this way, the experience of being food insecure may precipitate mental health problems. Equally, having a long-term mental health problem can lead to a drop in income because of labour market disadvantages [39], including being unable to work due to ill health and structural barriers such as stigma, and therefore could subsequently lead to food insecurity. Theoretically, the social security system should act as a safety net to protect the incomes of people living in poverty and those unable to work due to ill health, but evidence suggests that current social security policies and processes in the UK, such as the role out of Universal Credit [40], may in fact be contributing to food bank use [41].

There is a dearth of literature on whether and how the relationship between food insecurity and mental health varies by ethnic group in the UK. In this study, we address this gap in the current literature by utilising population-based data to

first explore the relationship between food insecurity and ethnicity and the relationship between food insecurity and mental health in the UK and second assess how the relationship between food insecurity and mental health in the UK differs by ethnic group. We explore food insecurity among respondents reporting a long-term mental health condition.

## Methods

### Study population and variables

The Family Resource Survey (FRS) [42] is a repeat cross-sectional survey which collects information on a representative sample of private households in the UK. Its primary purpose is to provide information to inform the development, monitoring and evaluation of social welfare policy. It provides annual statistics and commentary on circumstances and income from all sources; housing tenure; caring needs and responsibilities; disability (including physical and mental health); pension participation; savings and investment; and self-employment. The 2019−20 FRS (published in 2021) included data for the first time on household food insecurity.

Households in the FRS are defined as one person living alone or a group of people who may not necessarily be related living at the same address and who share cooking facilities and a living space, such as a sitting, dining, or living room. Within each household is a "Household Reference Person" (HRP), who is defined as the highest income householder, and households may include one or more "benefit units" (families); the head of the benefit unit may not necessarily be the HRP. After consultation with the Office for National Statistics, analyses were conducted for the HRP member of the household (n = 19,210). Theoretically the HRP, if leading a traditional household (rather than a household of multiple single non-relative cohabitees such as in a flat share, etc.), may shoulder the mental burden of being the highest earner and ensuring food security.

The study uses publicly available ONS secondary data. This data, which is in the public domain, is fully anonymised; access and use of this data poses no ethical risk. The data is available for public use via GOV.UK. Ethical consent was not needed for this study, however the broader research programme, of which this study is part, received ethical approval from the University of York Health Sciences Research Governance Committee on 23rd November 2020 (Reference: HSRGC/2020/418/F).

### Sociodemographic characteristics

Age of the household reference person (HRP) was collected in 10-year age range categories from age 16 through age 85 and over. Ethnicity was collapsed from five categories (White; Mixed/multiple; Asian/Asian British; Black/African/Caribbean/Black British; Other ethnic group) into four categories (White; Asian/Asian British; Black/African/Caribbean/Black British; Mixed/multiple/other) to increase sample size for regression models. "Other" ethnic group incorporated the FRS ethnicity categories "Mixed/ Multiple ethnic groups" and "Other ethnic group". Cohabitation was defined as "married/cohabitation" if the respondent was married, in a civil partnership, or cohabiting. The HRP was coded to "single/divorced/separated/widowed" if they responded they were single, widowed, divorced, had a civil partnership dissolved, or were separated. Housing tenure was categorised to "owned" if the HRP owns their home outright or owns with a mortgage, to "rented from council/housing association" if the HRP rents from either source, or to "privately rented" if the respondent rents privately, whether furnished or unfurnished. Household occupancy was determined by summing the number of adults and the number of dependent children within the household. The HRP was asked whether they received any state benefits in their own right, including Universal Credit, Housing Benefit, Working Tax Credit, Child Tax Credit, Income Support, Jobseeker's Allowance, Employment and Support Allowance, Carer's Allowance, or any/more than one of these; they were categorised as not receiving any benefits if they replied negatively or "none" when asked about each benefit in turn, and categorised as receiving benefits if they responded affirmatively to receiving any benefit.

## Household food security

Household food insecurity was assessed using the 10-item United States Department of Agriculture Adult Food Security Survey module [43]. One person was identified by the interviewer as the person with the best information about the food preparation and shopping for the household; this person was chosen to respond to the food security questions for each household. To ensure continuity between responses to the food insecurity questions and other questions in the FRS, food insecurity questions related to a 30-day rather than a 12-month reference period, as used to monitor food insecurity in the US, Canada and worldwide. Analysis suggests that use of the 30-day reference period likely under-estimates annual food insecurity prevalence [44].

The household food security questionnaire assesses quantitative and qualitative aspects of access to food and food supply and intake, including anxiety or perceived inadequacy of food intake or supply access, and hunger. Each of the questions had affirmative ("yes", "often true", "sometimes true", "3 or more" days) and negative ("no", "never true", "2 or fewer" days) responses; each affirmative response was scored a value of 1 and the responses summed across the questions to generate a final score range of 0–10. Households were then categorised into four categories of food security: (1) high food security (score = 0); (2) marginal food security (score = 1 or 2); (3) low food security (score = 3–5); (4) very low food security (score = 6–10). Households with high or marginal food security were considered "food secure" while those with low to very low food security were considered "food insecure".

## Health outcomes

Adult Participants, including the HRP, were queried about their health in the FRS to assess their wider sense of well-being beyond material and financial circumstances. Participants were asked whether they had any physical or mental health conditions or illnesses lasting or expecting to last for 12 months or more. The question was asked in two stages allow the survey to assess the presence of an illness lasting 12 months of or more and then to differentiate between different conditions. All adults were asked: Do you have any physical or mental health conditions or illnesses lasting or expected to last for 12 months or more? Respondents who answered 'yes' to this question were then asked whether this affected their physical or mental health. In this study, responses were coded to "yes" if the respondent answered responded affirmatively to their conditions lasting 12 months or more affecting their mental health, "no" if they responded negatively or with "don't know". Participants who chose not to respond were coded to missing. Among participants who responded affirmatively, they were additionally asked whether any of these conditions affected their mental health (categorised to "yes" or "no"). The question relating to mental health conditions lasting or expected to last 12 months or more does not refer to any particular diagnosis but does give an indication of mental health problems that are likely to be longer in duration and may for some people be classified as a disability. The variable also has the benefit of enabling self-report which although not clinician/diagnosis led, does ensure the experiences of respondents who may be experiencing distress but may not want to disclose a particular diagnosis, or alternatively may be awaiting assessment and treatment are included. This is particularly important for our sample given that people from ethnic minority backgrounds may be more likely to experience difficulty accessing mental health services to receive a diagnosis or support [27]. The duration provides some confidence that the experiences described by respondents are more than transient/short term issues.

## Statistical analysis

All analyses were conducted in R version 4.0.2 (R Foundation for Statistical Computing, Vienna, Austria, 2020). As mentioned above, analyses were conducted for the HRP member of the household (n = 19,210). Participant characteristics were described using number (n) and percentage (%) for categorical measures or median and interquartile range (IQR) (reported as a single number) for continuous measures due to non-Normal data. Characteristics were examined by food security status and their differences assessed using Kruskal-Wallis tests. We used unadjusted and adjusted logistic regression and estimated marginal means were used to examine the relationship between food security status and

longstanding illness affecting mental health. Regression models were run separately for different ethnic groups (Table 2 and Table 3). A directed Acyclic Diagram (DAG) (see S1 Fig) was drawn to assist in the selection of covariates for inclusion in the adjusted models; confounding variables adjusted for in the models include age, ethnicity, receipt of benefits, cohabitation, income, household occupancy, and housing tenure. Missing data were handled using a complete case analysis.

## Results

### Sample characteristics

Table 1 outlines the demographic characteristics of the sample. Overall, there were fewer female than male HRP respondents (41.7% compared to 58.3%). The majority of the sample identified as White (90.7%); 4.8% reported their ethnicity as Asian/Asian British and 2.5% were from a Black, African, Caribbean or Black British background. Two percent of HRP respondents described themselves as being from a mixed or "other" ethnic background. Around two thirds of the sample (64.6%) owned their home outright or with a mortgage. A fifth (20.1%) reported being in receipt of social security payments and the mean household income was £610 (£364-£ [1,32]) per week, before housing costs. Less than half of HRPs responded as having a longstanding illness (44%) and just over a fifth of those respondents (22.3%) reported that their longstanding illness affected their mental health.

92.3% of HRPs lived in households classified as food secure in the past thirty days, with 86.7% reporting high and 5.6% reporting marginal food security. Almost 8% were food insecure (7.8%) with 3.9% reporting low food security and very low food security in their household.

### Demographic characteristics by food security status, ethnicity and mental health

We observed differences across all demographic characteristics by food security status with the exception of household occupancy (Table 2).

Households with younger HRPs were more likely to experience food insecurity, with almost two-thirds of HRPs aged 25–54 (63.8%), more likely to have a HRP identifying as non-White (15.6% compared to 8.8%), alongside being more likely to be female (61.5% compared to 40%), and single, divorced, widowed, or separated (73.7% compared to 41.6%). HRPs in food insecure households were less financially secure compared to those who were food secure, reporting a lower income, being less likely to own their own home, and more likely to be in receipt of any benefits. A higher proportion also reported a longstanding illness which affected their mental health (58.3% compared to 17.8%).

Food insecurity was associated with increased odds of having a longstanding illness affecting mental health among HRPs from all ethnic groups (Table 3) (unadjusted: Odds Ratio (OR) 6.46 (5.60, 7.45); adjusted: 2.01 (1.76, 2.39)).

In multivariate analysis (see DAG in Supporting information for selection of covariates), HRPs reporting household food insecurity in the past 30 days and identifying as Asian/Asian British reported the highest odds of having a longstanding illness affecting mental health (OR=2.63 (1.05, 6.56)) followed by those identifying as White (OR=2.05 (1.72, 2.45)).

## Discussion

These analyses augment growing evidence on UK food insecurity, identifying demographic differences in food insecurity and in its relationship with mental health. In our study, experiences of household food insecurity in the past 30 days were associated with unstable circumstances such as being younger, single, divorced or widowed, renting, having a lower income, and being in receipt of benefits. Aligning with existing evidence [14], we identified significant ethnic differences in food insecurity – notably, and notwithstanding the different sample sizes in each group, 20% of Black/African/Caribbean/Black British HRP respondents are food insecure compared to 7% of White British respondents.

**Table 1. Characteristics of the head of household.**

| | Total n=19210 | |
|---|---|---|
| | **N** | **Median[IQR]/%** |
| **Sex** | | |
| Female | 8066 | 41.7 |
| Male | 11204 | 58.3 |
| **Age** | | |
| 16-24 | 451 | 2.3 |
| 25-34 | 2300 | 12 |
| 35-44 | 3109 | 16.2 |
| 45-54 | 3352 | 17.4 |
| 55-59 | 1797 | 9.4 |
| 60-64 | 1689 | 8.8 |
| 65-74 | 3528 | 18.4 |
| 75-84 | 2984 | 15.5 |
| 85+ | | |
| **Ethnicity** | | |
| White | 17422 | 90.7 |
| Asian/Asian British | 929 | 4.8 |
| Black/African/Caribbean/Black British | 475 | 2.5 |
| Mixed/multiple/other | 384 | 2 |
| **Marital/cohabitation** | | |
| Married/civil partnership/cohabitation | 10738 | 55.9 |
| Single/divorced/widowed/separated | 8472 | 44.1 |
| **Tenure** | | |
| Owned | 12404 | 64.6 |
| Privately rented | 3262 | 17 |
| Rented from council/housing association | 3544 | 18.4 |
| **Household occupancy** | 19210 | 2 (1,3) |
| **In receipt of benefits** | | |
| No | 15353 | 79.9 |
| Yes | 3857 | 20.1 |
| **Household income** | 19210 | 610 (364, 1032) |
| **Food security** | | |
| High | 16651 | 86.7 |
| Marginal | 1069 | 5.6 |
| Low | 741 | 3.9 |
| Very low | 749 | 3.9 |
| **Longstanding illness affecting mental health** | | |
| No | 6593 | 77.7 |
| Yes | 1890 | 22.3 |

We identified a relationship between food insecurity and a select mental health indicator, specifically between food insecurity and reporting a long-standing health condition affecting mental health (12 months or more). Stratifying by ethnicity, we found that the degree of worse mental health reported is greater for some minority ethnic groups.

**Table 2. Distribution of characteristics by food security status.**

| | Food insecure n=1490 (7.8%) | | Food secure n=17720 (92.2%) | | |
|---|---|---|---|---|---|
| | N | Median[IQR]/% | N | Median[IQR]/% | p-value* |
| **Sex** | | | | | <0.001 |
| Female | 916 | 61.5 | 7090 | 40 | |
| Male | 574 | 38.5 | 10630 | 60 | |
| **Age** | | | | | <0.001 |
| 16-24 | 73 | 4.9 | 378 | 2.1 | |
| 25-34 | 286 | 19.2 | 2014 | 11.4 | |
| 35-44 | 378 | 25.4 | 2731 | 15.4 | |
| 45-54 | 353 | 23.7 | 2999 | 16.9 | |
| 55-59 | 149 | 10 | 1648 | 9.3 | |
| 60-64 | 113 | 7.6 | 1576 | 8.9 | |
| 65-74 | 105 | 7 | 3423 | 19.3 | |
| 75-84 | 33 | 2.2 | 2951 | 16.7 | |
| 85+ | | | | | |
| **Ethnicity** | | | | | <0.001 |
| White | 1257 | 84.4 | 16165 | 91.2 | |
| Asian/Asian British | 84 | 5.6 | 845 | 4.8 | |
| Black/African Caribbean/Black British | 95 | 6.4 | 380 | 2.1 | |
| Mixed/multiple/other | 54 | 3.6 | 330 | 1.9 | |
| **Cohabitation** | | | | | <0.001 |
| Married/civil partnership/cohabitation | 392 | 26.3 | 10346 | 58.4 | |
| Single/divorced/widowed/separated | 1098 | 73.7 | 7374 | 41.6 | |
| **Tenure** | | | | | <0.001 |
| Owned | 253 | 17 | 12151 | 68.6 | |
| Privately rented | 398 | 26.7 | 2864 | 16.2 | |
| Rented from council/housing association | 839 | 56.3 | 2705 | 15.3 | |
| **Household occupancy** | 1490 | 2 (1,3) | 17720 | 2 (1,3) | 0.6 |
| **In receipt of benefits** | | | | | <0.001 |
| No | 460 | 30.9 | 14893 | 84 | |
| Yes | 1030 | 69.1 | 2827 | 16 | |
| **Household income** | 1490 | 376 (242, 542) | 17720 | 642 (380, 1070) | <0.001 |
| **Long-standing illness affecting mental health** | | | | | <0.001 |
| No | 393 | 41.7 | 6200 | 82.2 | |
| Yes | 549 | 58.3 | 1341 | 17.8 | |

*Chi-squared or Kruskal-Wallis test for differences between food security status

Food insecure Asian/Asian British HRP respondents had higher odds of reporting a long-term mental health condition than other ethnic groups reporting food insecurity. It is well established that Asian/Asian British populations in the UK experience multiple barriers to mental health service use and, as a consequence, uptake of primary care mental health services are low. Barriers include experiences and expectations of racial mistreatment, difficulties in aligning religious and Western biomedical belief systems, and community-level stigma towards mental illness [45,46]. It is possible that poor access to services for Asian/Asian British people exacerbates the relationship between food insecurity and mental illness, particularly where mental illness lasts 12 months or more (as is the case with this outcome).

**Table 3. Logistic regression results for the association between food insecurity (outcome) and longstanding illness affecting mental health, stratified by ethnicity.** Unadjusted and adjusted odds ratios (OR) and 95% confidence intervals (CI) for the association between reporting a longstanding illness affecting mental health and household food insecurity (vs. food security) among Household Reference Persons (HRPs). Adjusted models control for age, benefit receipt, cohabitation, income, household occupancy, and housing tenure.

| | Unadjusted | | | | Adjusted* | | | |
|---|---|---|---|---|---|---|---|---|
| | N | β/OR | 95% CI | p-value | N | β/OR | 95% CI | p-value |
| **All ethnic groups (Outcome: Food secure vs. food insecure)** | | | | | | | | |
| Longstanding illness affecting mental health | 8483 | 6.46 | (5.60, 7.45) | <0.001 | 8483 | 2.01 | (1.70, 2.39) | <0.001 |
| **White** | | | | | | | | |
| Longstanding illness affecting mental health | 7942 | 7.15 | (6.15, 8.33) | <0.001 | 7942 | 2.05 | (1.72, 2.45) | <0.001 |
| **Asian/Asian British** | | | | | | | | |
| Longstanding illness affecting mental health | 274 | 4.44 | (2.06, 9.46) | <0.001 | 274 | 2.63 | (1.05, 6.56) | 0.04 |
| **Black/African/Caribbean/Black British** | | | | | | | | |
| Longstanding illness affecting mental health | 143 | 2.21 | (0.89, 5.43) | 0.08 | 143 | 0.83 | (0.26, 2.57) | 0.75 |
| **Mixed/multiple/other** | | | | | | | | |
| Longstanding illness affecting mental health | 124 | 3.58 | (1.44, 9.05) | 0.006 | 124 | 1.12 | (0.29, 4.14) | 0.86 |

*adjusted for age, benefit receipt, cohabitation, income, household occupancy, housing tenure

Evidence suggests that both food insecurity [18] and mental health [47] can be perceived as shameful and stigmatised among Asian/Asian British communities, which can lead to difficulties remaining unacknowledged and services being avoided [48]. It is possible both that food insecurity and long-standing mental health conditions remain under/unreported exacerbating the link between the two, and that Asian/Asian British people experiencing food insecurity are at high risk of mental health and simultaneously are more likely than other ethnic groups to be ostracised from familial and social support, placing them at risk of isolation, financial hardship and worsening mental health.

While the relationship between long-term health conditions related to mental health and food insecurity was strongest among Asian/Asian British respondents, it was quite similar for those who identify as White. It is possible that some of the factors cited to explain this relationship among Asian/Asian British respondents are also applicable to White respondents, in particular poor access to mental health services (albeit not racially motivated) and familial and social isolation.

Considering the outcome measure as a marker of long-term disability illuminates the potential structural mechanisms between food insecurity and long-term mental health conditions. There is an established relationship between disability and severe food insecurity [49] and there is a significant over-representation of disabled people referred to food banks [50]. Qualitative research finds that food insecurity and food bank use among disabled people is at least in part driven by structural issues associated with the disability social system. This includes, low awareness of benefit entitlements and eligibility, long waits, incorrect decisions, and often stressful application and assessment process which are damaging to people's mental and physical health and deepen exclusion [50].

## Strengths and limitations

This is one of the first studies to consider ethnic differences in the relationship between food insecurity and mental health in the UK population. As such, it makes a significant contribution to the evidence on UK food insecurity, highlighting the complex relationship between food insecurity, ethnicity and health. The study is, nevertheless, subject to limitations.

On the advice of the Office for National Statistics (ONS), the analysis was conducted using the Household Reference Person (HRP), who is the householder with the highest income or, where two people in the household have the same income, the older of the two. The HRP was the chosen participant from the FRS; this choice was made because the HRP was theorised to shoulder the mental burden of being the highest earner and ensuing food insecurity (a 'household-level'

variable). It is also worth noting that The Family Resources Survey (FRS) collects benefit receipt information separately for each adult in the household. In our analysis, we followed the advice of the Office for National Statistics and focused on the HRP for consistency and to align with other household-level variables. However, this may underreport household benefit receipt if only non-HRP members are recipients, say a partner or older relative, most applicable to benefits like Pension Credit. This limitation may have caused some misclassification of economic disadvantage.

As noted above, individual level interviews are conducted with all adults in participant FRS households and data at the individual level are also available for all adults in FRS households. An alternative research design could have been to use individual-level data for all adults in the household, applying the household-level food insecurity data to all adults in the household, while adjusting for clustering effects by household. This approach, which was not adopted here due to advice from the ONS and our own theoretical position on food insecurity as a 'household-level' variable. However, this is an important avenue for further research, one which would provide a larger sample for looking at how individual level characteristics in the household, including ethnicity, are related to household-level food insecurity and allow for insights into the mental health of other household members, themselves perhaps dependent on the incomes of others, such as the HRP, in the household and occupying a household position of lesser power or agency.

The use of the HRP in our analyses of food insecurity likely explains the lower number of female respondents (41.7% compared to 51% in the general population) and possibly also the lower number of minority ethnic respondents (90.7% of the sample identified as White compared to 81.7% in the general population), with implications for food insecurity reporting given the known higher prevalence of food insecurity among ethnic minorities and the well-established role of mothers in shouldering the burden of food insecurity within the household (REFS)

Households were considered to be "food insecure" if they reported low to very low food security while households who reported marginal food security were categorised as "food secure"; there is some debate as to whether marginal food security is a reflection of food insecurity rather than food security and categorising the variables in this way misrepresents the realities/scale of food insecurity in the population. The food insecurity questions apply to a 30-day rather than a 12-month reference period, likely further contributing to an underestimation of food insecurity [35]. The cross-sectional nature of the data makes it impossible to conclude the direction of the relationship between food insecurity and mental health. It is worth noting that the Understanding Society dataset does allow for some longitudinal analysis of food insecurity. Assessing the direction of the relationship is additionally complicated by the variable time periods of the food insecurity and long-standing illness questions: the length of the longstanding illness (12 months) implies that it could potentially start before and hence be a precursor to any food insecurity (30 days). Assessment of mental health was highly circumscribed by available data to assess mental health in the FRS, which was seen as the best dataset to employ for analyses of food insecurity; the long-standing illness affecting mental health variable could feasibly capture people whose physical illness is impacting their mental health condition rather than reflecting a mental health condition per se. Nevertheless, the question on long-standing health does have advantages in not requiring the person to have already had the health condition for 12 months at the time of interview, meaning the questions should pick up people with acute as well as chronic mental ill health and, while some people may potentially be missed out, for example, if they do not have a specific diagnosis and consequently do not consider the question to be relevant to them, the question is broad enough in its framing to allow those with and without a formal mental health diagnosis to answer affirmatively.

Future research could apply (quasi) experimental approaches to get closer to causal estimates and consider the use of clinical mental health scales to assess current mental health.

## Conclusion

Using data from 2019/20, this study finds a strong relationship between food insecurity and specific indicators of mental health among all ethnic groups. Since this data was collected, there have been sharp increases in both food insecurity and mental illness in the UK, first as a consequence of the Covid-19 pandemic and then as a result of rapid rises in the

cost of food, energy and housing, rendering the cost of essentials increasingly expensive while incomes stagnate. Our findings are likely to be an underestimate of current levels of food security and thus the potential consequences for mental health today may be greater than we have been able to ascertain using 2019/20 data. Policy makers should urgently address rising food insecurity among all groups while simultaneously employing targeted and culturally appropriate interventions to tackle ethnic inequalities in both food insecurity and mental health. Improving the UK social security system to provide a stable and adequate source of income for low income households, and those who are unable to work due to long-standing health conditions, could help to target the entrenched and harmful relationship between poverty, food insecurity and mental ill health.

## Supporting information

**S1 Fig. Directed Acyclic Graph (DAG) food insecurity and long-term health condition affected by mental health.** (PDF)

## Author contributions

**Conceptualization:** Maddy Power, Katie Pybus.

**Data curation:** Tiffany Yang.

**Formal analysis:** Maddy Power, Katie Pybus, Benham Tajik.

**Funding acquisition:** Maddy Power.

**Investigation:** Tiffany Yang.

**Methodology:** Maddy Power.

**Project administration:** Maddy Power.

**Supervision:** Maddy Power.

**Writing – original draft:** Maddy Power, Tiffany Yang, Katie Pybus.

**Writing – review & editing:** Maddy Power, Tiffany Yang, Katie Pybus, Benham Tajik.

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
