## [Decision Letter · Decision Letter 0]

28 May 2025

PONE-D-25-06411Association between food insecurity, ethnicity, and mental health in the UK: An analysis of the Family Resource SurveyPLOS ONE

Dear Dr. Power,

Thank you for submitting your manuscript to PLOS ONE. After careful consideration, we feel that it has merit but does not fully meet PLOS ONE’s publication criteria as it currently stands. Therefore, we invite you to submit a revised version of the manuscript that addresses the points raised during the review process.

We look forward to receiving your revised manuscript.

Kind regards,

Emily Lowthian

Academic Editor

PLOS ONE

Journal Requirements:

“This research was funded by a Research Fellowship in Humanities and Social Sciences held by Maddy Power. Grant Number: 221021/Z/20/Z”

Reviewers' comments:

Reviewer's Responses to Questions

**Comments to the Author**

1. Is the manuscript technically sound, and do the data support the conclusions?

Reviewer #1: Yes

Reviewer #2: Yes

2. Has the statistical analysis been performed appropriately and rigorously? 

Reviewer #1: No

Reviewer #2: Yes

3. Have the authors made all data underlying the findings in their manuscript fully available?

Reviewer #1: Yes

Reviewer #2: Yes

4. Is the manuscript presented in an intelligible fashion and written in standard English?

Reviewer #1: Yes

Reviewer #2: Yes

5. Review Comments to the Author

Reviewer #1: Immediately, this is obvious to researchers familiar with UK household food insecurity research that should have been completed/reported on previously so the merit of this analysis is clear. This builds on the authors' previous work well.

There are some good potential contributions here, however, the use of the anxiety question from the day before and the statistical methods are problematic. With some revision to focus on the ethnicity and long term illness only, this can still make a good contribution to the literature and I hope the authors will find my comments useful.

The abstract structure is different from usual PLoS ONE.

In the introduction, bring in the more recent data on FI from the 2024 report (noting the 2021 report appears later), which puts it at 10% (https://www.gov.uk/government/statistics/united-kingdom-food-security-report-2024/united-kingdom-food-security-report-2024-theme-4-food-security-at-household-level#household-food-security-status). Good to see the link to Food Foundation's work.

To build the literature up further, please include a link to the systematic review )https://pmc.ncbi.nlm.nih.gov/articles/PMC10200655/pdf/S136898001900435Xa.pdf) and a paper exploring the demographics associated with FI (doi:10.1017/S1368980020000087). I am less comfortable to see food bank use applied as a proxy for very low FI (line 67) given how relatively few people who are FI use food banks - again as shown in the updated Govt 2024 report. The discussion around how there is a two way relationship between IF and mental health was good to read. This recent systematic review about FI and severe mental illness may also help to extend the literature (https://onlinelibrary.wiley.com/doi/full/10.1111/jpm.12969).

Further regarding ethnicity, is it a consideration that minority ethnic groups may be less likely to report FI and mental health in both cases due to cultural expectations? Perhaps this was picked up in your earlier work in Bradford which has been referenced. So rather than kinship networks mitigating FI, it can also be 1) less awareness of the support available such as food banks or other aid 2) cultural barriers to seeking support (stigma, as with many other populations – this does appear in the discussion); 3) issues with the food available if they do seek help being inappropriate or unfamiliar. Many years ago Donkin and Dowler wrote about this issue in the context of food deserts. It would be beneficial to outline the possible mechanisms.

While I very much agree, wholeheartedly, that current welfare/social policies in the UK contribute to food bank use, please do not use a Trussell Trust report to substantiate this. Have a look at Milbourne's recent work on more than food banks, Loopstra’s work on the impact of the UC roll out on food insecurity, or Williams and May's genealogy of food banks to use academic sources. As you note elsewhere, when there was the £20 uplift on UC and furlough, there were much lower rates of FI. Further, Trussell Trust does not represent all food banks so their data will be incomplete.

Overall in the intro, mental health as a label needs to be specified. When discussing mental health for the analysis, please clarify if the mental health being assessed here is common mental disorders (typical) only for the longer term conditions, then only anxiety for recent experiences. Also add in the population prevalence of the relevant mental health outcome as you have done for FI.

Methods:

Overall there are two substantial concerns: the use and coding of the anxiety variable to identify mental health issues and the use of linear regression.

Hopefully in the discussion it will be noted that all data used are cross section so causation cannot be inferred from this analysis (yes, lines 375-6). Understanding Society did briefly collect data on FI.

Limitations around the FRS data collection are well described.

Is there any report to show how often the HRP may not claim benefits, but someone else in the household does? (lines 172-177). What about Pension Credit?

As the section progresses, there are more concerning issues with the analysis plans, variable choice and coding.

Is the anxiety question validated elsewhere (in the limitations, it states no)? This does not sound like the typical measures used to assess anxiety, such as GAD 7. Only asking how someone felt the day before is not adequate to identify a mental health outcome; the GHD-12 and GAD 7, or Wemwebs, ask about the previous 2 weeks and are validated. I realise you are working with what is available in the FRS.

There are issues with the anxiety variable and coding of it, which is unclear as described (lines 200-210). If the scale is 0-10, this is not what would be considered a continuous variable, but an ordinal variable as there are only 11 possible answers.

Re-reading, I think that longstanding mental illness is someone who said ‘yes’ to conditions lasting 12 months or more. However, in table 1 there is not reporting of longstanding illness on its own, only where it impacts on mental health. Looking at the table, anxiety is relatively low if the median is 2 with an IQR of 0- 5.

The statistical approaches are difficult to understand.

Food security is coded into four groups, then collapsed into two: food secure or insecure. Why use linear regression for (now) categorical outcome variables? However reading this again I think the linear regression was only for anxiety. The Kruskal-Wallis tests is sensible for categorical variables, but linear regression assumes a normal distribution of data, or it needs to be transformed, and linear variables for the outcome or dummy coding for categorical predictor variables. Binary logistic regression would be an appropriate choice, as has been used in table 3 where there are obvious odds ratios. Please make it clear in the table that the outcome is food insecure.

The beta coefficients are very high, and look more like ORs. Typically they range between -1 and 1. It would be better to recode anxiety into above and below the mean or median and re-run as a logistic regression model.

As it is written, this analysis is not appropriate for the research aims. The anxiety variable needs to be shown to be appropriate for use to identify anxiety (how does this coding compare to GHQ-12 for common mental disorders, for example?

Or, the paper would be sufficient without the anxiety data included. The issue of minority ethnic food insecurity, and long term illness, is enough.

Discussion:

The discussion brings in a lot of good literature to describe the results and picks up on some points I raise about the introduction. Long term mental health outcomes are discussed specifically (lines 312-313; 329-334) but I cannot see that they were measured sperate to long term physical health condition. The question asks if the LLTI is affecting mental health, this is different to long term mental health problems. I did not see where the data were collected to show that specifically mental health problems lasted 12 months or more (line 320).

Reviewer #2: A beautifully written article. The study utilises the 2019/20 Family Resource Survey. This enhances the credibility of the findings and made use of a large sample size of households for generalisation to a broader population. The focus on ethnicity provides a nuanced understanding of how food insecurity impacts mental health across diverse groups, something I know the lead author has specific expertise in.

I think this article is excellent, really well written and well-articulated. I have not found any computational or discussion errors, or significant issues with this. In fact, I have really enjoyed reading it. The authors should be extremely proud of this argument. My review below is mainly about minor areas of clarification.

• I think the discussion and focus on HRP and anxiety was absolutely fascinating and an absolute pleasure (and interest) to read, really well and clearly articulated too (145-157).

• I really liked the discussion about Sociodemographics (159-177). However, as food insecurity deals with issues of enough income, I was left wondering - What about either Before Housing Costs or After Housing Costs? And were these considered? If not, does this change the discussion?

• 227-228 – perhaps a note and a reference to Sociological literature about the gendered difference in HRP

• 241 – following reading your data here, I’m now wondering about the rising cost of living and the recent fluctuations in inflation. Is there something to discuss about the relative cost of food vs incomes here?

• 353 – whilst I agree that the HRP may well shoulder the burden of financial anxiety associated with earning enough and being able to feed the household, I also wonder about those who do not have this power, or seem to be powerless in this, i.e. those without a financial ability (not responsibility), perhaps those with lower earnings and do not make financial decisions – do they suffer with more anxiety through a lack of power? Could the authors add a sentence or two about those who are dependent on the ability of others, as they have less say and may feel even more anxious (if not, perhaps, totally excluded from being able to make food/financial decisions).

o I note that discussions happen around this in the following sentences (354-366), so perhaps you have covered this, but there is an interesting discussion to have about HRP and gender.

• 369-376 – I also tend to agree with what you hint at here. Perhaps you are either food secure, or food insecure (and no shades of grey) as having levels of (in)security may not be that beneficial to understanding the concept. Is it more binary than current accounts allow for?

6. PLOS authors have the option to publish the peer review history of their article (what does this mean? ). If published, this will include your full peer review and any attached files.

**Do you want your identity to be public for this peer review?** For information about this choice, including consent withdrawal, please see our Privacy Policy .

Reviewer #1: No

Reviewer #2: **Yes: ** Dave Beck

---

## [Author Response · Author response to Decision Letter 1]

2 Aug 2025

Please note that we have also included this text as an attachment.

Journal Requirements:

This has been checked and corrected as necessary.

“This research was funded by a Research Fellowship in Humanities and Social Sciences held by Maddy Power. Grant Number: 221021/Z/20/Z”

This statement has been added to the cover letter.

The following text has been added to the Methods section:

The study uses publicly available ONS secondary data. This data, which is in the public domain, is fully anonymised; access and use of this data poses no ethical risk. The data is available for public use via GOV.UK. Ethical consent was not needed for this study, however the broader research programme, of which this study is part, received ethical approval from the University of York Health Sciences Research Governance Committee on 23rd November 2020 (Reference: HSRGC/2020/418/F).

Review Comments to the Author

Reviewer #1: Immediately, this is obvious to researchers familiar with UK household food insecurity research that should have been completed/reported on previously so the merit of this analysis is clear. This builds on the authors' previous work well. There are some good potential contributions here, however, the use of the anxiety question from the day before and the statistical methods are problematic. With some revision to focus on the ethnicity and long term illness only, this can still make a good contribution to the literature and I hope the authors will find my comments useful.

Comment: Many thanks for your very helpful review of this article. We firmly agree with your comments on the anxiety variable and therefore have removed it entirely from the study. We hope this streamlines and strengthens the article to make it suitable for publication.

The abstract structure is different from usual PLoS ONE.

Comment: The abstract has been rewritten in line with journal format and guidelines.

In the introduction, bring in the more recent data on FI from the 2024 report (noting the 2021 report appears later), which puts it at 10% (https://www.gov.uk/government/statistics/united-kingdom-food-security-report-2024/united-kingdom-food-security-report-2024-theme-4-food-security-at-household-level#household-food-security-status). Good to see the link to Food Foundation's work.

The data has been updated.

To build the literature up further, please include a link to the systematic review )https://pmc.ncbi.nlm.nih.gov/articles/PMC10200655/pdf/S136898001900435Xa.pdf) and a paper exploring the demographics associated with FI (doi:10.1017/S1368980020000087).

Many thanks for the recommendations of this reference; this as now been added.

I am less comfortable to see food bank use applied as a proxy for very low FI (line 67) given how relatively few people who are FI use food banks - again as shown in the updated Govt 2024 report.

We very much agree that food bank use is a poor proxy for low food insecurity and this reference has now been removed.

The discussion around how there is a two way relationship between IF and mental health was good to read. This recent systematic review about FI and severe mental illness may also help to extend the literature (https://onlinelibrary.wiley.com/doi/full/10.1111/jpm.12969).

Many thanks for this suggestion, this reference has now been added.

Further regarding ethnicity, is it a consideration that minority ethnic groups may be less likely to report FI and mental health in both cases due to cultural expectations? Perhaps this was picked up in your earlier work in Bradford which has been referenced. So rather than kinship networks mitigating FI, it can also be 1) less awareness of the support available such as food banks or other aid 2) cultural barriers to seeking support (stigma, as with many other populations – this does appear in the discussion); 3) issues with the food available if they do seek help being inappropriate or unfamiliar. Many years ago Donkin and Dowler wrote about this issue in the context of food deserts. It would be beneficial to outline the possible mechanisms.

A discussion of possible mechanisms has been added, which draws closely on work completed over the last 10 years by this team.

While I very much agree, wholeheartedly, that current welfare/social policies in the UK contribute to food bank use, please do not use a Trussell Trust report to substantiate this. Have a look at Milbourne's recent work on more than food banks, Loopstra’s work on the impact of the UC roll out on food insecurity, or Williams and May's genealogy of food banks to use academic sources. As you note elsewhere, when there was the £20 uplift on UC and furlough, there were much lower rates of FI. Further, Trussell Trust does not represent all food banks so their data will be incomplete.

We completely agree with the issues surrounding the use of the Trussell Trust reference; this has now been removed and these additional references added – many thanks for the recommendations.

Overall in the intro, mental health as a label needs to be specified. When discussing mental health for the analysis, please clarify if the mental health being assessed here is common mental disorders (typical) only for the longer term conditions, then only anxiety for recent experiences. Also add in the population prevalence of the relevant mental health outcome as you have done for FI.

The following text has been added to the introduction:

Like food insecurity, mental health is increasingly becoming an urgent and widespread public health concern in the UK and worldwide. According to the World Health Organization, a mental disorder is characterized by a clinically significant disturbance in an individual’s cognition, emotional regulation, or behaviour. It is usually associated with distress or impairment in important areas of functioning. There are many different types of mental disorders. Mental disorders may also be referred to as mental health conditions. The latter is a broader term covering mental disorders, psychosocial disabilities and (other) mental states associated with significant distress, impairment in functioning, or risk of self-harm. In 2019, 1 in every 8 people, or 970 million people around the world were living with a mental disorder, with anxiety and depressive disorders the most common.

This text is now included in the methods:

The question relating to mental health conditions lasting or expected to last 12 months or more does not refer to any particular diagnosis but does give an indication of mental health problems that are likely to be longer in duration and may for some people be classified as a disability. The variable also has the benefit of enabling self-report which although not clinician/diagnosis led, does ensure the experiences of respondents who may be experiencing distress but may not want to disclose a particular diagnosis, or alternatively may be awaiting assessment and treatment are included. This is particularly important for our sample given that people from ethnic minority backgrounds may be more likely to experience difficulty accessing mental health services to receive a diagnosis or support. The duration provides some confidence that the experiences described by respondents are more than transient/short term issues.

Methods:

Overall there are two substantial concerns: the use and coding of the anxiety variable to identify mental health issues and the use of linear regression.

Hopefully in the discussion it will be noted that all data used are cross section so causation cannot be inferred from this analysis (yes, lines 375-6). Understanding Society did briefly collect data on FI.

In addition to noting the cross-sectional nature of the data in the discussion/limitation we have also added the following sentence: ‘It is worth noting that the Understanding Society dataset does allow for some longitudinal analysis of food insecurity.’

Limitations around the FRS data collection are well described.

Is there any report to show how often the HRP may not claim benefits, but someone else in the household does? (lines 172-177). What about Pension Credit?

As far as we are aware, there is no specific FRS publication that quantifies how often benefits are received by non-HRP household members, and therefore we have acknowledged that this scenario is likely, particularly in multi-generational or cohabiting households by adding a discussion of this in our limitations. Specifically:

It is also worth noting that The Family Resources Survey (FRS) collects benefit receipt information separately for each adult in the household. In our analysis, we followed the advice of the Office for National Statistics and focused on the HRP for consistency and to align with other household-level variables. However, this may underreport household benefit receipt if only non-HRP members are recipients, say a partner or older relative, most applicable to benefits like Pension Credit. This limitation may have caused some misclassification of economic disadvantage.

As the section progresses, there are more concerning issues with the analysis plans, variable choice and coding. Is the anxiety question validated elsewhere (in the limitations, it states no)? This does not sound like the typical measures used to assess anxiety, such as GAD 7. Only asking how someone felt the day before is not adequate to identify a mental health outcome; the GHD-12 and GAD 7, or Wemwebs, ask about the previous 2 weeks and are validated. I realise you are working with what is available in the FRS.

Because of the considerable limitations of the anxiety variable this has now been removed from the analysis entirely. This major change the study also addresses the following comments:

There are issues with the anxiety variable and coding of it, which is unclear as described (lines 200-210). If the scale is 0-10, this is not what would be considered a continuous variable, but an ordinal variable as there are only 11 possible answers.

Re-reading, I think that longstanding mental illness is someone who said ‘yes’ to conditions lasting 12 months or more. However, in table 1 there is not reporting of longstanding illness on its own, only where it impacts on mental health. Looking at the table, anxiety is relatively low if the median is 2 with an IQR of 0- 5.

To clarify the question on long-standing illness, we have added the following text in the methodology section: The question was asked in two stages to allow the survey to assess the presence of an illness lasting 12 months of or more and then to differentiate between different conditions. All adults were asked: Do you have any physical or mental health conditions or illnesses lasting or expected to last for 12 months or more? Respondents who answered ‘yes’ to this question were then asked whether this affected their physical or mental health. In this study, responses were coded to “yes” if the respondent answered responded affirmatively to their conditions lasting 12 months or more affecting their mental health, “no” if they responded negatively or with “don’t know”. Participants who chose not to respond were coded to missing.

The following text has been added to the limitations:

Assessment of mental health was highly circumscribed by available data to assess mental health in the FRS, which was seen as the best dataset to employ for analyses of food insecurity; the long-standing illness affecting mental health variable could feasibly capture people whose physical illness is impacting their mental health condition rather than reflecting a mental health condition per se. Nevertheless, the question on long-standing health does have advantages in not requiring the person to have already had the health condition for 12 months at the time of interview, meaning the questions should pick up people with acute as well as chronic mental ill health and, while some people may potentially be missed out, for example, if they do not have a specific diagnosis and consequently do not consider the question to be relevant to them, the question is broad enough in its framing to allow those with and without a formal mental health diagnosis to answer affirmatively.

The statistical approaches are difficult to understand. Food security is coded into four groups, then collapsed into two: food secure or insecure. Why use linear regression for (now) categorical outcome variables? However reading this again I think the linear regression was only for anxiety. The Kruskal-Wallis tests is sensible for categorical variables, but linear regression assumes a normal distribution of data, or it needs to be transformed, and linear variables for the outcome or dummy coding for categorical predictor variables. Binary logistic regression would be an appropriate choice, as has been used in table 3 where there are obvious odds ratios.

The text relating to this part of the analysis has now been removed with the removal of the anxiety variable.

Please make it clear in the table that the outcome is food insecure.

We have revised the caption in Table 3, added a note to the table and updated the outcome row label in Table 3 to make these clearer.

The beta coefficients are very high, and look more like ORs. Typically they range between -1 and 1. It would be better to recode anxiety into above and below the mean or median and re-run as a logistic regression model.

The anxiety variable has now been removed given its limitations. The removal of the anxiety variable also addresses the two comments below.

As it is written, this analysis is not appropriate for the research aims. The anxiety variable needs to be shown to be appropriate for use to identify anxiety (how does this coding compare to GHQ-12 for common mental disorders, for example?

Or, the paper would be sufficient without the anxiety data included. The issue of minority ethnic food insecurity, and long term illness, is enough.

Discussion:

The discussion brings in a lot of good literature to describe the results and picks up on some points I raise about the introduction. Long term mental health outcomes are discussed specifically (lines 312-313; 329-334) but I cannot see that they were measured sperate to long term physical health condition. The question asks if the LLTI is affecting mental health, this is different to long term mental health problems.

More information on the long-term mental health outcome, including its separate measurement from long-term physical health, is now provided in the methodology section and considered in the limitations.

I did not see where the data were collected to show that specifically mental health problems lasted 12 months or more (line 320).

More detail on this question an

---

## [Decision Letter · Decision Letter 1]

4 Sep 2025

Association between food insecurity, ethnicity, and mental health in the UK: An analysis of the Family Resource Survey

PONE-D-25-06411R1

Dear Dr. Power,

We’re pleased to inform you that your manuscript has been judged scientifically suitable for publication and will be formally accepted for publication once it meets all outstanding technical requirements.

Kind regards,

Emily Lowthian

Academic Editor

PLOS ONE

Additional Editor Comments (optional):

Reviewers' comments:

Reviewer's Responses to Questions

**Comments to the Author**

1. If the authors have adequately addressed your comments raised in a previous round of review and you feel that this manuscript is now acceptable for publication, you may indicate that here to bypass the “Comments to the Author” section, enter your conflict of interest statement in the “Confidential to Editor” section, and submit your "Accept" recommendation.

Reviewer #1: (No Response)

Reviewer #2: All comments have been addressed

2. Is the manuscript technically sound, and do the data support the conclusions?

Reviewer #1: Yes

Reviewer #2: Yes

3. Has the statistical analysis been performed appropriately and rigorously? 

Reviewer #1: Yes

Reviewer #2: Yes

4. Have the authors made all data underlying the findings in their manuscript fully available?

Reviewer #1: Yes

Reviewer #2: Yes

5. Is the manuscript presented in an intelligible fashion and written in standard English?

Reviewer #1: Yes

Reviewer #2: Yes

6. Review Comments to the Author

Reviewer #1: Thank you for the taking the time to address the reveiw comments thoroughly, I can appreciate the effort involved. The paper will be a good contribution to the literature, well done.

Reviewer #2: Thanks for the update. All comments that I made have been met. Still really liked this article, I think it was really well written and I added some minor comments, which have also now been met.

7. PLOS authors have the option to publish the peer review history of their article (what does this mean? ). If published, this will include your full peer review and any attached files.

**Do you want your identity to be public for this peer review?** For information about this choice, including consent withdrawal, please see our Privacy Policy .

Reviewer #1: No

Reviewer #2: **Yes: ** Dave Beck

---

## [Editor Report · Acceptance letter]

PONE-D-25-06411R1

PLOS ONE

Dear Dr. Power,

I'm pleased to inform you that your manuscript has been deemed suitable for publication in PLOS ONE. Congratulations! Your manuscript is now being handed over to our production team.

Kind regards,

on behalf of

Dr. Emily Lowthian

Academic Editor

PLOS ONE